# Phylogeny and Flow Cytometry of the Genus *Kalidium* Moq. (Amaranthaceae s.l.) in Kazakhstan

**DOI:** 10.3390/plants12142619

**Published:** 2023-07-11

**Authors:** B. B. Osmonali, P. V. Vesselova, G. M. Kudabayeva, M. V. Skaptsov, A. I. Shmakov, N. Friesen

**Affiliations:** 1Institute of Botany and Phytointroduction, Almaty 050040, Kazakhstan; be96ka_kz@mail.ru (B.B.O.); pol_ves@mail.ru (P.V.V.);; 2Department of Biodiversity and Bioresources, Faculty of Biology and Biotechnology, Al-Farabi Kazakh National University, Almaty 050040, Kazakhstan; 3South Siberian Botanical Garden, Faculty of Biology, Altai State University, Barnaul 656906, Russia; mr.skaptsov@mail.ru (M.V.S.); alex_shmakov@mail.ru (A.I.S.); 4Botanical Garden, School of Biology/Chemistry, Osnabruck University, 49076 Osnabruck, Germany

**Keywords:** Chenopodiaceae, Salicornioideae, Salicornieae, Kazakhstan, hybrid, tetraploid

## Abstract

The article presents data on phylogeny, genome size, and ploidy of species of the genus *Kalidium* Moq. in the flora of Kazakhstan. Genus *Kalidium* belongs to the tribe Salicornieae of the subfamily Salicornioideae of the family Chenopodiaceae and unites eight species, the main range of which covers the Iranian-Turanian and Central Asian deserts. There are four species in the flora of Kazakhstan: *K. foliatum*, *K. caspicum*, *K. schrenkianum*, and the recently described *K. juniperinum*. Populations of species of the genus *Kalidium* in the saline deserts of Kazakhstan occupy large areas, often forming monodominant communities. Sometimes there is a joint growth of two and very rarely three species of the genus. During the period of fieldwork (2021–2022), populations were identified in which these species grew together with a predominance, in most cases, of *K. caspicum*. Samples of representatives from 15 populations were collected for research. Selected plant samples were studied by flow cytometry to determine plant ploidy. Sequencing of nrITS and two chloroplast fragments were used to build a phylogenetic tree, including sequences from the NCBI database., A phylogenetic tree of species of the genus *Kalidium* was compiled, which takes previously published data into consideration. In the valley of the middle reaches of the Syrdarya River, tetraploid populations of *K. caspicum* were found. A hybrid between *K. foliatum* and *K. caspicum* was found in the Ili River valley (Almaty region, Uigur district). To identify phylogenetic processes at the intraspecific level, the SCoT (Start codon targeted) fingerprinting method was used.

## 1. Introduction

The genus *Kalidium* Moq. belongs to the tribe Salicornieae, whose subfamily Salicornioideae [1,2] is part of the largest family, Chenopodiaceae (Amaranthaceae s. lato), and is to be found in the North Turan deserts (Kazakhstan) [3]. All species of the genus *Kalidium* are euhalophytes [4] or halo-succulents, which are halophytes with relatively fleshy, succulent stems or leaves [5,6]. Until recently, the genus was divided into six species: *Kalidium foliatum* (Pall.) Moq., *K. caspicum* (L.) Ung.-Sternb., *K. cuspidatum* (Ung.-Sternb.) Grubov, *K. gracile* Fenzl, *K. schrenkianum* Bunge and *K. wagenitzii* (Aellen) Freitag and G. Kadereit. In 2020, the subspecies *K. cuspidatum* var. *sinicum* A. J. Li [7] was reclassified to species *K. sinicum* (A. J. Li) by H.C. Fu and Z.Y. Chu [8], and at the end of 2022, an eighth species of this genus, *K. juniperinum* Sukhor. and Lomon [9], was added. The main range of the genus covers the Irano–Turanian and Central Asian deserts [1,10].

Members of the genus *Kalidium* are small or dwarf glabrous shrubs, mostly with reduced, semi-amplex leaf laminae. The species *K. foliatum*, *K. wagenitzii*, and *K. juniperinum* are found to have roll-shaped (terete) and succulent leaf laminae, which can measure up to 1.2 cm in length [9]. The peduncles consist of three submerged flowers with one or two stamens and united segments with four or five teeth. The fruit has a parenchymatous pericarp and a thin, yellow, or brownish seed coat with a fine papillary surface [9].

Species of the genus *Kalidium* play an important role in maintaining the balance of grassland ecosystems and preventing soil erosion [11]. Comparative studies have shown that, as the dominant species in desert areas, species of the genus are highly tolerant to saline and alkaline soils as well as to drought [11,12]. As succulents, they are mainly used as winter fodder for camels, horses, and sheep [13]. In addition, Wang and Jia [14] showed that ethanol crude extracts of aerial parts of *K. foliatum* have high antibacterial properties.

A review of the literature has shown that most of the scientific work relating to the study of the *Kalidium* species is aimed at investigating the topical phenomenon of salt tolerance. Among the species particularly well-adapted to saltwater habitats are *K. foliatum* and *K. caspicum* [14,15,16,17,18,19,20,21].

A few works are related to studies of the genus’s systematics. Some papers contain information on the scope of the tribe and subfamily, in which two tribes (Halopeplideae and Salicornieae) are combined into one: Salicornieae [1,2]. A recently published paper by Chinese scientists provides information on the DNA barcoding of species of the genus *Kalidium* and substantiates the independence of the species *K. sinicum* [8].

This paper presents the results of our study—also at the molecular genetic level—of species within the genus *Kalidium* from Kazakhstan, the results of which facilitated comparative analysis with other regions of the world.

For the species *K. foliatum*, a genome-wide analysis of chloroplast DNA has already been performed, which showed that it is phylogenetically related to two species of Salicornia, *S. bigelovii* Torr. and *S. brachiata* Roxb. [22]. These molecular data support the taxonomic interpretation that all three species belong to the same tribe, Salicornieae. The resulting data from Wang et al. [22] research provided a new genetic resource for evolutionary and comparative genomic analysis with other Chenopodiaceae species. Currently, *Salicornia* includes annual plants with opposite pairs of fused leaves and bracts, three-flowered, in which lateral flowers touch below the central flower, a conduit embryo, and no perisperm. Distributed throughout the world (except Australia), it comprises 13–17 species. *Salicornia arabica* L. and *S. caspica* L. have been synonymized with *Kalidium caspicum*, and *S. europaea* var. *fruticosa* L. was recognized as a separate species of *Sarcocornia*, *S. fruticosa* (L.) A. J. Scott. [2].

In most cases, species of the genus *Kalidium* are used as an outgroup for various research works on other genera of the tribe Salicornieae [23,24,25]. There are also papers on the morphology of pollen from the Salicornieae tribe, including classification [26].

According to the data in the literature, the genus *Kalidium* is generally represented by three species in Kazakhstan: *K. caspicum*, *K. foliatum*, and *K. schrenkianum* [27,28,29,30,31,32]. However, in 2022, A.P. Sukhorukov and M.N. Lomonosova described a new species: *K. juniperinum* Sukhor. and Lomon., which occur mostly in the central and northern parts of Kazakhstan [9]. Unfortunately, this work was published after we had conducted our molecular genetic analysis, and as a consequence, we did not analyze *K. juniperinum* material. In addition, present studies are aimed at territories where species of the genus *Kalidium* grow abundantly, and K. *juniperinum* is found more rarely and to the north. Moreover, it is difficult to identify it morphologically as *K. juniperinum*, as the possibility of confusion with *K. foliatum* is very high. Furthermore, the species has not been confirmed genetically.

The relevance of this study lies in the fact that populations of *Kalidium* species in the saline deserts of Kazakhstan occupy large areas, as very few other plants are able to grow under such conditions. It should be noted that species of this genus form monodominant communities in most cases, with only occasional cooccurrences of two, or, very rarely, three species. Two species cannot usually dominate together, except under special circumstances where there is exposure to external, usually anthropogenic, influences. Additionally, according to our observations, two species that coexist together will hybridize. However, given the specific morphological structure of the *Kalidium* species, it is virtually impossible to visually identify hybrid plants in the field. Moreover, the hybrid form may not be the first generation, and the possibility of reverse hybridization with one of the parents cannot be excluded, in which characteristics of one of the parents are more pronounced: in our case, this is *K. foliatum*.

In the desert area of the Syrdarya valley, the dominant species of the genus *Kalidium* that form large communities include *K. caspicum* and *K. foliatum*. During the fieldwork period (2021–2022), populations were identified in which both species occurred, with *K. caspicum* predominating in most cases. Monopopulations dominated only by *K. caspicum* and covering a large area were also observed. In these populations, individuals of *K. caspicum* were well developed and attained a larger size in comparison to other populations but this did not differ morphologically.

The original plant samples were studied using flow cytometry, in particular, these were the species *K. caspicum* and *K. foliatum*, as these are the most common and also form large populations. One of the most common uses of this method is to study hybridogenic processes that manifest as polyploidy and aneuploidy [33]. The discovery of such hybrid and polyploid specimens in our samples necessitated additional analyses. It was decided to apply molecular genetic techniques to analyze internal transcribed spacer (ITS) and chloroplast fragments in order to more accurately identify hybrid and polyploid samples.

One of the most important molecular methods in the study of relatedness in supraspecific systematics is the comparison and analysis of aligned DNA sequences of individual genome fragments and plant plastomes. Analysis of ITS ribosomal DNA (ITS nrDNA) is the most popular for genome fragment studies (nuclear DNA), while the plastome uses a wide range of genes and introns [34].

## 2. Results

As a result of expeditionary trips, samples of *Kalidium* representatives from 15 populations were collected for the research. This article, however, presents the results of flow cytometry studies of the samples examined in only 11 populations.

Based on a summary and critical analysis of the data we obtained during our studies, including sequences from the NCBI database using QGIS 2.14 software (https://qgis.org, accessed on December 2022), a point distribution map of the samples studied was produced (Figure 1 and Appendix A).

### 2.1. Flow Cytometry

Genome size (DNA content in nuclei) was determined in two species of the genus *Kalidium* (*K. caspicum* and *K. foliatum*) from 11 populations. The results, shown in Table 1 and Table 2 and in Appendix A, were checked against the Chromosome Counts Database (CCDB).

The results obtained by flow cytometry showed the presence of polyploid (tetraploid) species (Table 1 and Table 2) in the following populations: B01, B08, B09, B10, and B11.

The study of the DNA content of diploid populations of *K. foliatum* and *K. caspicum* is complicated by possible hybrids and backcross hybrids with intermediate morphology and DNA content (smeared genome). In addition, many single samples showed intermediate DNA content between *K. foliatum* and *K. caspicum*, especially in populations B03, B07 (Table 1—putative hybrids). Intermediate DNA content apparently characterizes hybrids of the first generation and backcross hybrids of the second or third generation. Examples of ungated histograms of *Kalidium* specimens examined are shown in Figure 2.

### 2.2. Molecular Phylogeny

Internal transcribed spacer (nrITS) and chloroplast (trnQ-rpS16 and trnL(UAG)-rpL32) fragments were sequenced for 15 populations of three species: *K. caspicum*, *K. foliatum*, and *K. schrenkianum*, as shown in Figure 3 and Figure 4. The ITS fragments for populations B10, B11, B12, and B13 were each made of two samples, and for B14 of three samples.

According to the ITS phylogenetic tree (Figure 3), the species of the genus *Kalidium* are divided into two large groups. The first group includes *K. foliatum*, *K. gracile*, *K. wagenitzii*, *K. sinicum (K. cuspidatum* var. *sinicum)*, and *K. cuspidatum* var. *cuspidatum*. The second group comprises the species *K. caspicum* and *K. schrenkianum* (Figure 3). Similarly, in the first group, the species *K. sinicum* clearly (100%) diverges from the other species, *K. foliatum*, *K. gracile*, and *K. cuspidatum*. Furthermore, among the three species mentioned above, bootstrap support was 83%, and between *K. cuspidatum* and *K. gracile* 73%.

Most of the *K. foliatum* specimens we studied (B02, B05, B06, and B13.2) were arranged quite predictably, although specimen B13 was positioned among the specimens of *K. caspicum*, relatively distant from *K. foliatum* in the ITS phylogenetic tree. Most taxa form clear monophyletic clades in the ITS tree, but some sequences from GenBank have a questionable position, namely accessions KU975203 *K. schrenkianum;* HM131637—*K. cuspidatum*; DQ340146—*Kalidiopsis wagenitzii*.

A combined plastid tree (trnQ-rps16 + rpl32-trnL) with only our own sequences and with *Halochnemum strobilaceus* and *Halostachys belangeriana* as the outgroup shows only two well-supported clades (Figure 4): *K. caspicum* including *K. schrenkianum* and *K. foliatum*. The caspicum clade is divided into three subclades with good support, more or less, by geographic origin. It is interesting that the plastid sequences of *K. schrenkianum* accession B15 are similar to sequences of three accessions of *K. caspicum* (B04, B14, and B12).

The following data were obtained by running the data through the JModeltest software: TVM + G, −lnL 2301.19646, AIC 4698.392920.

### 2.3. SCoT Results

For SCoT analysis, 3 samples were collected from each population. Thus, with a total of 15 populations studied, a total of 45 samples were analyzed. Initially, 10 SCoT primers were used: SCoT2, SCoT4, SCoT11, SCoT12, SCoT13, SCoT14, SCoT16, SCoT17, SCoT21, SCoT23. However, only 6 primers gave good results: SCoT11, SCoT12, SCoT13, SCoT14, SCoT21, SCoT23 (Appendix A). A UPGMA tree of the three species of the genus *Kalidium* was constructed from the resulting matrix using the MEGA 7.0 software (Figure 5). In the UPGMA tree with SCoT matrix data, the accessions are clearly classified by origin. In addition, SCoT analysis of the UPGMA tree showed some variation in the proximity data for samples B10.1, B11.1, and B12.1. Thus, it appears that samples B10.1 and B11.1 are close to the B09 population and B12.1 to the B11 population.

The three accessions of *K. schrenkianum* are classified as a sister group to *K. foliated*. Furthermore, in the PCA analysis based on ScoT data (Appendix A), species are clearly separated.

## 3. Discussion

According to the phylogenetic tree (ITS), the species of the genus *Kalidium* are divided into two large groups. The first group includes *K. foliatum*, *K. gracile*, *K. wagenitzii*, *K. sinicum (K. cuspidatum* var. *sinicum)*, and *K. cuspidatum* var. *cuspidatum*. The second group comprises the species *K. caspicum* and *K. schrenkianum* (Figure 3). These figures are in good agreement with those cited by Liang and Wu [13]. Similarly, in the first group, the species *K. sinicum* clearly (100%) diverges from the other species, *K. foliatum*, *K. gracile*, and *K. cuspidatum*. Furthermore, among the three species mentioned above, bootstrap support was 83%, and between *K. cuspidatum* and *K. gracile* 73%. These values are close to those obtained by Liang and Wu [13]. Two varieties, *K. cuspidatum* var. *sinicum* and *K. cuspidatum* var. *cuspidatum* were previously identified within *K. cuspidatum*. Morphological features were presented in the Flora of China in 2003 [43]. The first of them is now considered an independent species *K. sinicum* Liang and Wu, in which the main distinguishing features are as follows: leaves 1–1.5 mm; densely branched plants [13,43]. Since the program assigned the specimens specified *K. cuspidatum* (MW725164, MW725165, DQ340148) in the group to *K. sinicum*, they most likely belong to *K. cuspidatum* var. *sinicum*, while samples of *K. cuspidatum* var. *cuspidatum* actually represent *K. cuspidatum*, in which the distinguishing features are as follows: leaves 1.5–3 mm; low-branched plants [43].

According to the ITS tree, *K. gracile* is closely related to *K. cuspidatum*. Unfortunately, we were not able to attain the herbarium specimen *K. schrenkianum* (KU975203) from the Kyzylorda deserts, which was found by H. Freitag and S. Rilke (26500) in the vicinity of Novokazalinsk (now Aitekebi) in the Kazaly district of Kazakhstan [1,2]. Therefore, we cannot explain why this specimen fell into the *K. gracile* species group.

We now turn to the results obtained for *K. foliatum*, which has the most extensive range of all species in the genus, for which, accordingly, several open questions related to its phylogeny remain. Most of the *K. foliatum* specimens we studied (B02, B05, B06, and B13.2) were arranged quite predictably, although specimen B13 was positioned among the specimens of *K. caspicum*, relatively distant from *K. foliatum* in the phylogenetic tree (ITS). However, it should be taken into consideration that our *K. foliatum* specimens were growing on the edge of a large population of *K. caspicum*, near a road slope where there was a change in microrelief.

To clarify the results obtained, we sequenced another sample of *K. foliatum* (13.2) from this population, which resulted in its placement in its own sample group. When the results of ITS and the chloroplast tree were compared (Figure 3 and Figure 4), B13 appeared to be in the *K. foliatum* group, revealing the hybrid nature of this sample.

Without further research, it is not possible to explain why only the *K. wagenitzii* and DQ340146 *Kalidiopsis wagenitz* collected by H. Freitag collection number 28801, which was also cited in several papers [1,2], became included in the *K. foliatum* sample group in the resulting phylogenetic tree. A detailed study of this issue is highly relevant, as *K. wagenitzii* is considered endemic to the territory of Turkey.

Another specimen, *K. cuspidatum* (HM131638), once used as an outgroup [24], fell into the *K. foliatum* group. Unfortunately, we did not have data on herbarium specimens for this study, but according to our assumption, this is a technical plucking when extracting DNA (co-dominance) or when loading data into the database. There is also an unlikely case of incorrect identification of the herbarium specimen; given the level of researchers, this option is the least suitable for explanation. In addition, in the research work in which this specimen was presented, it does not play a particularly important role in the phylogeny studied, since it was treated like an Outgroup. In the second group of species, *K. caspicum* and *K. schrenkianum*, analysis of specimens of *K. schrenkianum* showed a standard result, with little intraspecific variation (ITS), and is most likely related to the geographical location of their habitat [13]. When the chloroplast DNA sequences were analyzed, the *K. schrenkianum* sample (B15) was found to be a member of the *K. caspicum* group (Figure 4). It should be noted, however, that the population of *K. schrenkianum* (which consisted of no more than 20 specimens) grew within a large population of *K. caspicum*. In this case, the lack of material on this species played its role, but given the identification of tetraploids and a hybrid for the most common species, it allows us to leave this issue for further research. On the other hand, only one specimen of *K. schrenkianum*, which, according to ITS, is close to the species *K. caspicum*, could simply be aligned to the characteristic closest species. Moreover, it is very unlikely, but cannot be excluded, that this circumstance suggests a hybrid origin of the *K. schrencianum* (B15) specimen. Unfortunately, we did not have enough material in our own collections and resources in the NCBI database to confirm or refute this version.

Analysis of *K. caspicum* by ITS fragments showed a standard arrangement in the phylogenetic tree (Figure 3), although analysis by chloroplast fragments showed that the species was distributed according to collection sites (Figure 4). Given that the specimens studied were collected from different areas located at different hypsometric heights, they differed quite well in the phylogenetic tree. Thus, samples of *K. caspicum* (B01, B03, B07, B08, B09, B10, and B11) collected in the Kyzylorda region grew at absolute altitudes between 60 m and 160 m above sea level. Samples B12 and B14, from the Almaty region, were found at altitudes between 530 m and 620 m, whereas B04, from the Zhetysu region, at an altitude of 1010 m.

The tetraploid specimens B01, B08, B09, and B10 were the most interesting. They were combined into one group (Table 2), proving the presence of polyploidization within *K. caspicum.* Only sample B11 did not show the expected result (not being in the tetraploid group), which we attribute to a possible technical error in a sample selection for analysis from this population.

When discussing the results of the SCoT analysis, in which *K. schrenkianum* was seen to stay close to *K. foliatum*, attention must be drawn to the fact that the phylogenetic tree data (ITS and chloroplast DNA) show this species to be close to *K. caspicum* (Figure 5).

Otherwise, all three species studied differed well from each other. Of the submitted specimens in population B07, all appeared to be identical, while in the other populations, small differences between the specimens studied were recorded.

The PCA show the interspecific arrangement of the species studied. They show, in particular, a clear distinction between the three species, proximity between *K. schrenkianum* and *K. foliatum*, and proximity between *K. foliatum* and *K. caspicum* (Appendix A).

A cross-population analysis of *K. caspicum* showed that populations B03, B07, and B08 were distinct (Appendix A). Samples of these populations were selected for research in the Kyzylorda region. The B03 and B07 populations were located much farther away from the other populations, namely in the Aral and Kazaly districts.

The geographical location of populations B03, B07, and B08 corresponds, more or less, to PCA 9A and 9B of the point map in Appendix A. Although Appendix A shows population B08 as distant from populations B03 and B07, the SCoT analysis histogram shows that it is closer to these populations than B09 and B10, on the other side of the Syrdarya.

Comparison of the map of population locations in general with data on the confinement of specifically tetraploid species indicates their concentration in the middle reaches of the Syrdarya River (within the Zhalagash and Zhanakorgan districts), where solonchak deserts are most widely represented. This may also be due to an increase in air temperature in the western direction.

Unfortunately, due to the small number of *K. foliatum* populations collected in nature and analyzed, we cannot yet reliably explain the results shown in the histograms and map (Appendix A).

## 4. Material and Methods

The objects of the study are species of the genus *Kalidium* growing in the territory of Kazakhstan: *K. caspicum*, *K. foliatum*, and *K. schrenkianum* (not considering the recently described species, *K. juniperinum* Sukhor. and Lomon.), as shown in Figure 6.

Classical botanical methods (route reconnaissance, ecological-systematic, and ecological-geographical) were used in the research process. The herbarium collection was carried out according to the method of Skvortsov [44], and fundamental summaries were used to identify the material collected: *Flora of Kazakhstan* [28], *Illustrated Plant Identifier of Kazakhstan* [29], and *Central Asian Plant Identifier* [30]. The names of plant species were taken from the International Plant Names Index (IPNI) database.

The material was collected within the Zhetysu, Almaty, and Kyzylorda regions of Kazakhstan. The largest amount of analyzed material was collected in Kyzylorda oblast, where the largest populations of the study objects occurred. All points of our collections, as well as those cited by other researchers in the National Center for Biotechnology Information (NCBI) database, are presented in Appendix A.

Unfortunately, the pattern obtained from the analysis of samples from four populations (including a sample of *K. schrenkianum* species from the Almaty region) did not allow it to be interpreted accordingly. We believe this is due to the accumulation of significant amounts of metabolites in the cells of these plant samples because of the late period (September) of their collection.

### 4.1. Flow Cytometry

The relative DNA content was determined by flow cytometry techniques using propidium iodide by internal standardization. The sample leaves, dried with silica gel, were ground using a blade in a 1 mL Tris-MgVl2 buffer solution with the composition 0.2 M tris base, 4 mM MgCl2, 0.5% Triton X-100, 50 µg/mL RNase, 50 µg/mL propidium iodide, pH 7.5 [45]. Fluorescence data of isolated nuclei were detected using a Cytoflex flow cytometer (Beckman Coulter, Brea, CA, USA) with a 488 nm laser source. Histograms were visualized and processed using CytExpert software (Beckman Coulter, USA). The standards *Glycine max* (L.) Merr. ‘Polanka’ 2C = 2.5 pg [46], *Petroselinum crispum* (Mill.) Fuss ‘Champion moss’ 2C = 4.46 pg [47], and *Pisum sativum* L. ‘Ctirad’ 2C = 9.09 pg [48] were used as internal standards for DNA content determination. DNA content (2C, pg) was calculated according to the following formula: 2C, pg (Sample Peak Average/Standard Peak Average) × 2C Standard.

The ploidy of the test samples was determined by external standardization without changing the cytometer settings, using the index of the difference between the mean peak values of the test species with known or suspected ploidy:Index = Sample Peak Average/Standard Peak Average

### 4.2. Molecular Genetics Methods

Extraction of DNA from leaves dried with silica gel was carried out using the NucleoSpin Plant II Mini kit (MACHEREY-NAGEL GmbH & Co. KG, Düren, Germany). The ITS fragment was amplified using ITS-A [49] and ITS-4 [50] primers. The chloroplast fragments trnQ-rpS16 and trnL (UAG)-rpL32 were amplified using the primers described by Shaw et al. [51]. The Polymerase Chain Reaction (PCR) mix consisted of 1 µL DNA, 1 µL primer, 10 µL Red HS Taq 2x Mix, and 8 µL distilled water.

### 4.3. Amplification and Sequencing

To perform the PCR, the amplification protocol for Red Mix was used, involving a 20 µL reaction mixture with 2× HS Taq Mix Red (Biozym Scientific GmbH, Hessisch Oldendorf, Germany), where the mix was 1 µL each of direct primer and reverse primer, 10 µL Red Mix, and 8 µL *H*_2_. The amplified products were tested by electrophoresis on a 1.5% agarose gel stained with ethidium bromide. The DNA fragments were visualized under UV light on a Gel I X20 Lmager (INTAS Science Imaging Instruments GmbH, Göttingen, Germany) and documented using a Mitsubishi P93D printer (Mitsubishi Elec. Corp., Chiyoda City, Japan). The PCR products were sent to Microsynth SeqLab (Göttingen, Germany; www.microsynth.seqlab.de) for sequencing. The sequences from all individuals were manually edited in Chromas Lite 2.1 (Technelysium Pty Ltd., South Brisbane, QLD, Australia) and aligned with ClustalX [52]; the alignment was manually corrected using MEGA7.0 [53].

### 4.4. Phylogenetic Analyses

Both datasets (nrITS and the cpDNA markers) were analyzed separately through Fitch parsimony with the heuristic search option in PAUP version 4.0 b10 [54] with MULTREES, TBR branch swapping, and 100 replicates of random addition sequence. Gaps were treated as missing data. The consistency index (CI) was calculated to estimate the amount of homoplasy in the character set [55]. The most parsimonious trees returned by the analysis were summarized in one consensus tree using the strict consensus method. Bootstrap support (BS) was performed using 1000 pseudoreplicates to assess the support of the clades [56]. Bayesian phylogenetic analyses were also performed using MrBayes 3.1.23 [57]. The sequence evolution model was chosen by following the Akaike information criterion (AIC) obtained from jModelTest2 [58]. Two independent analyses with four Markov chains were run for 10 million generations, sampling trees every 100 generations. The first 25% of the trees were discarded as burn-in. The remaining 150,000 trees were combined into a single dataset and a majority-rule consensus tree was obtained along with posterior probabilities (PP).

Two species from the tribe Salicornieae were selected as an outgroup: *Halocnemum strobilaceum* (Pall.) M. Bieb. and *Halostachys belangeriana* (Moq.) Botsch.

### 4.5. The Start Codon Targeted (SCoT) Method

To assess the genetic polymorphism, population samples of the genus *Kalidium* were tested with 10 SCoT primers, resulting in the selection of six primers showing polymorphism for further analysis.

The PCR was performed in a Professional Thermocycler (Biometra, Germany), with the following program: pre-precipitation—01:30 min at 94 °C, then 36° cycles (00:45 min—+94 °C, 00:45 min—+50 °C, 1:30 min—+72 °C), and the final step—6:00 min.—+72 °C and 90:00 min at 12 °C. The DNA was separated in an electrophoresis chamber using agarose gels with an agarose concentration of 1.5% in a TVE buffer using ethidium bromide. The duration of electrophoresis was 3.5–4 h at an electric field voltage of 85 V. The DNA was visualized using INTAS Science Imaging with Intas GDS software [59].

A 100 bp-DNA EXTENDED ladder was used as a DNA standard. The electrophoresis results were analyzed by the presence (1) or absence (0) of bands in the gel, followed by matrix generation. IBM SPSS Statistics was used to carry out PCA analysis of the data. A dendrogram showing the degree of similarity between the populations studied and the genetic distance was constructed using MEGA7.0 software [53]. To do this, the numeric values 1 and 0 in the matrix were replaced by alphabetic values (1 to A and 0 to G), and the sample names were formatted in Fasta format. For the matrix of SCoT data, see Appendix A.

The matrix of SCoT data was also analyzed in the SPSS program (Version 28 https://www.ibm.com/products/spss-statistics accessed on 20 December 2022). A PCA analysis was executed with this program. The PCA is based on a correlation matrix of characters (Appendix A) using the Pearson correlation coefficient.

## 5. Conclusions

As a result of field studies of the phytocenotic role of *Kalidium* species in the composition of vegetation of the Kyzylorda region, it was found that among the halophytic plant communities confined to the Syrdarya River valley, phytocenoses with the participation of *Kalidium* species are widely represented. At the same time, it is *K. caspicum* that acts as the dominant species. The analysis of samples of *K. caspicum* taken in the middle course of Syrdarya River (Zhanakorgan and Shili districts) by flow cytometry revealed tetraploid populations occupying considerable space and differing by the larger size of plants.

At the same time, *K. foliatum*, both in the Syrdarya River valley and in the Ili River valley, are found much less frequently, and *K. schrenkianum* in the Syrdarya River valley does not occur at all, occupying more eastern territories.

According to the phylogenetic tree of species of the genus *Kalidium* compiled (taking into account the NCBI database):-The species of the genus *Kalidium* are divided into two large groups. The first group: *K. foliatum*, *K. gracile*, *K. wagenitzii*, *K. sinicum*, and *K. cuspidatum*. The second group: *K. caspicum* and *K. schrenkianum*. A promising direction for further study of the genus is additional research on the isolation of superspecific categories.-Specimens previously attributed to *Kalidium* cuspidatum variations (*K. cuspidatum* var. *sinicum* and *K. cuspidatum* var. *cuspidatum*) represent two independent taxa: *K. cuspidatum* var. *sinicum*, understood as *K. sinicum* and *K. cuspidatum* var. *cuspidatum*, as *K. cuspidatum*.

A specimen of hybrid (between *K. foliatum* and *K. caspicum*) origin was found in the Ili River valley, according to molecular genetic studies.

## Figures and Tables

**Figure 1 plants-12-02619-f001:**
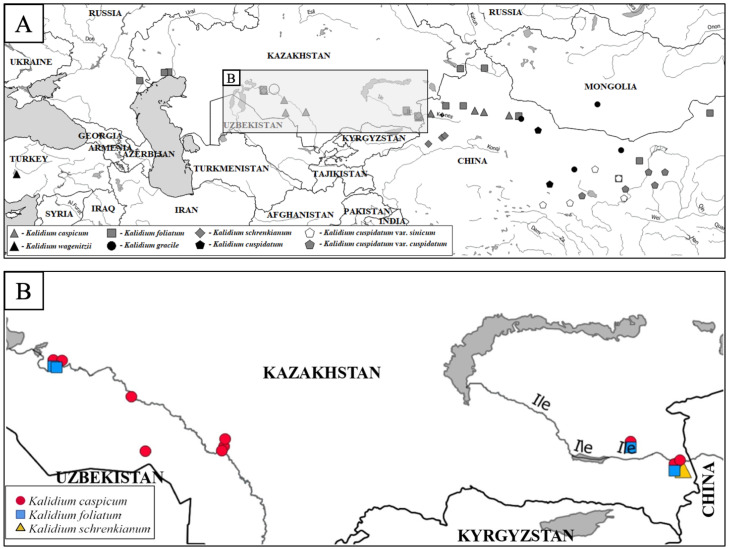
Distribution map of the samples examined. (**A**) All samples presented in the work; (**B**) Samples collected by us.

**Figure 2 plants-12-02619-f002:**
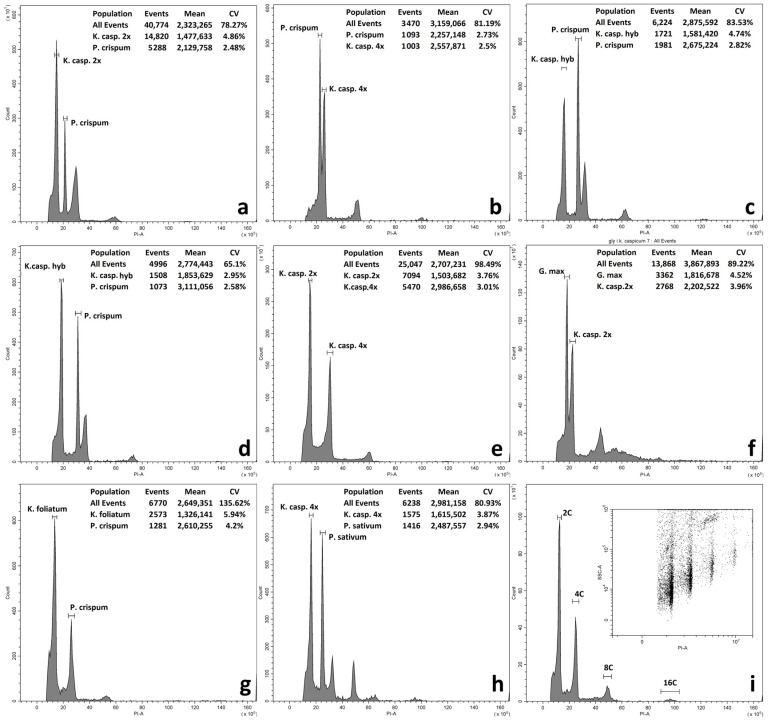
Examples of ungated histograms of *Kalidium* specimens examined. (**a**)—diploid specimen *K. capsicum* and standard *P. crispum*; (**b**)—tetraploid specimen *K. caspicum* and standard *P. crispum*; (**c**,**d**)—putative hybrids of *K. caspicum* × *K. foliatum* B07 and B03 pop. (2C = 2.616 and 2.663, respectively); (**e**)—di- and tetraploid *K. caspicum* combined; (**f**)—diploid *K. capsicum* and *G. max* standard; (**g**)—*K*. *foliatum* and *P. crispum*; (**h**)—tetraploid specimen *K*. *caspicum* and standard *P*. *sativum*; (**i**)—example of endopolyploidy of *K*. *caspicum*.

**Figure 3 plants-12-02619-f003:**
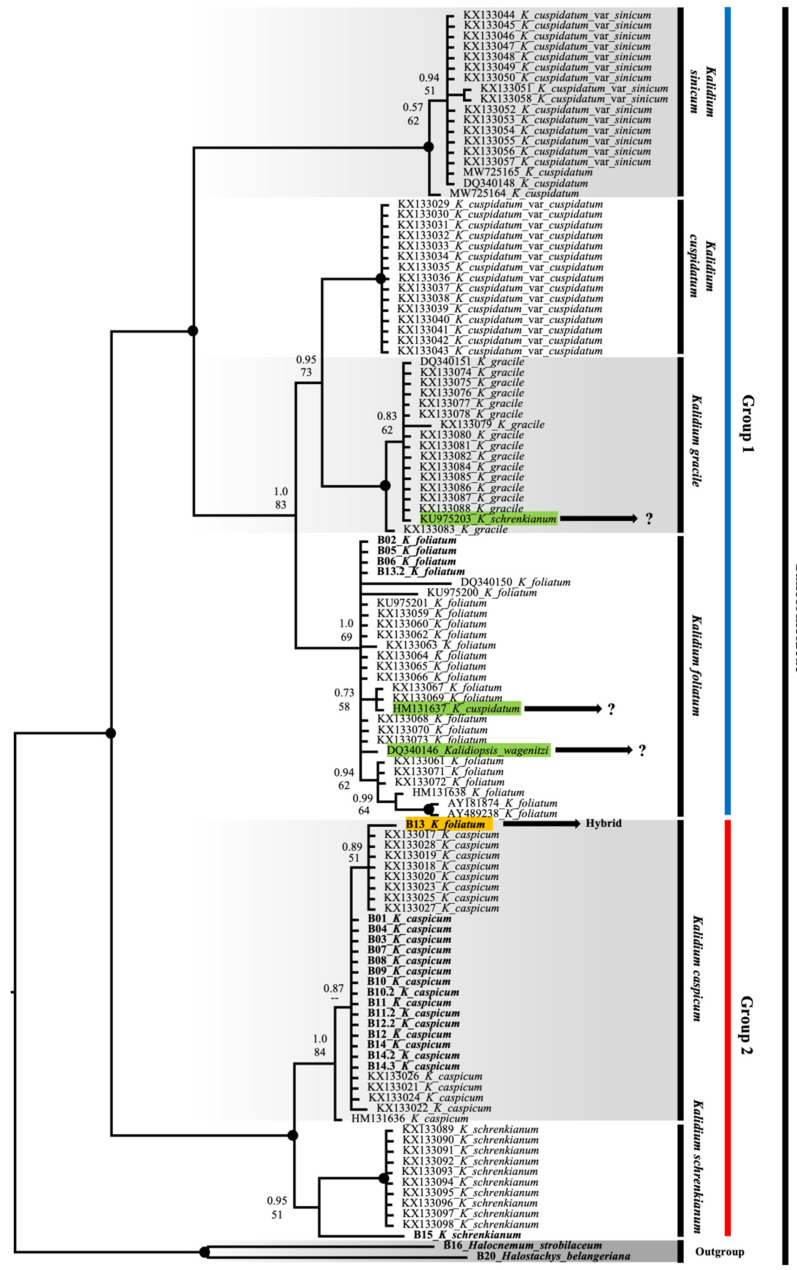
ITS tree of the genus *Kalidium*. The joint presence of Bayesian probability greater than 0.98 and bootstrap support greater than 95% is indicated by a black dot. The putative Cuspidatum group is marked in blue and the Caspicum group in red. For visual convenience, the species boundaries are highlighted in grey. The specimens we studied are shown in bold. Hybrid B13 *K. foliatum* is marked in yellow. Samples with unclear locations in the tree are highlighted in green. When running the data through the JModeltest program, the following data was obtained: GTR + G, −lnL 1871.40999, AIC 4232.819980.

**Figure 4 plants-12-02619-f004:**
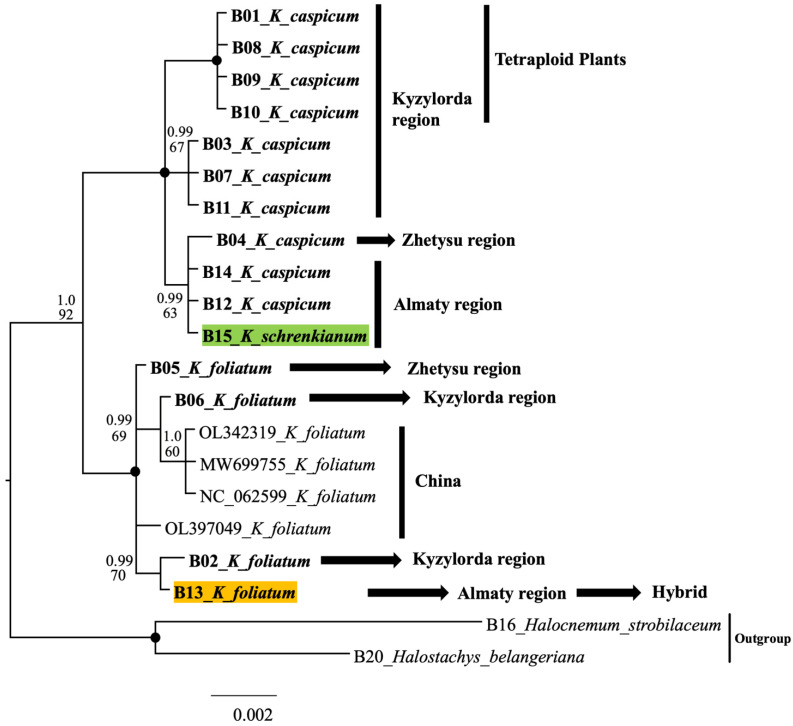
Plastiden Tree (trnQ-rps16 and rpl32-trnL) of the genus *Kalidium*. The joint presence of Bayesian probability greater than 0.98 and bootstrap support greater than 95% is indicated by a black dot. Bold type indicates the specimens that we have isolated. Hybrid B13 *K. foliatum* is marked in yellow. Specimen B15 of *K. schrenkianum*, whose position in the tree is not clear, is highlighted in green. The arrows and lines show the areas from where these samples were taken.

**Figure 5 plants-12-02619-f005:**
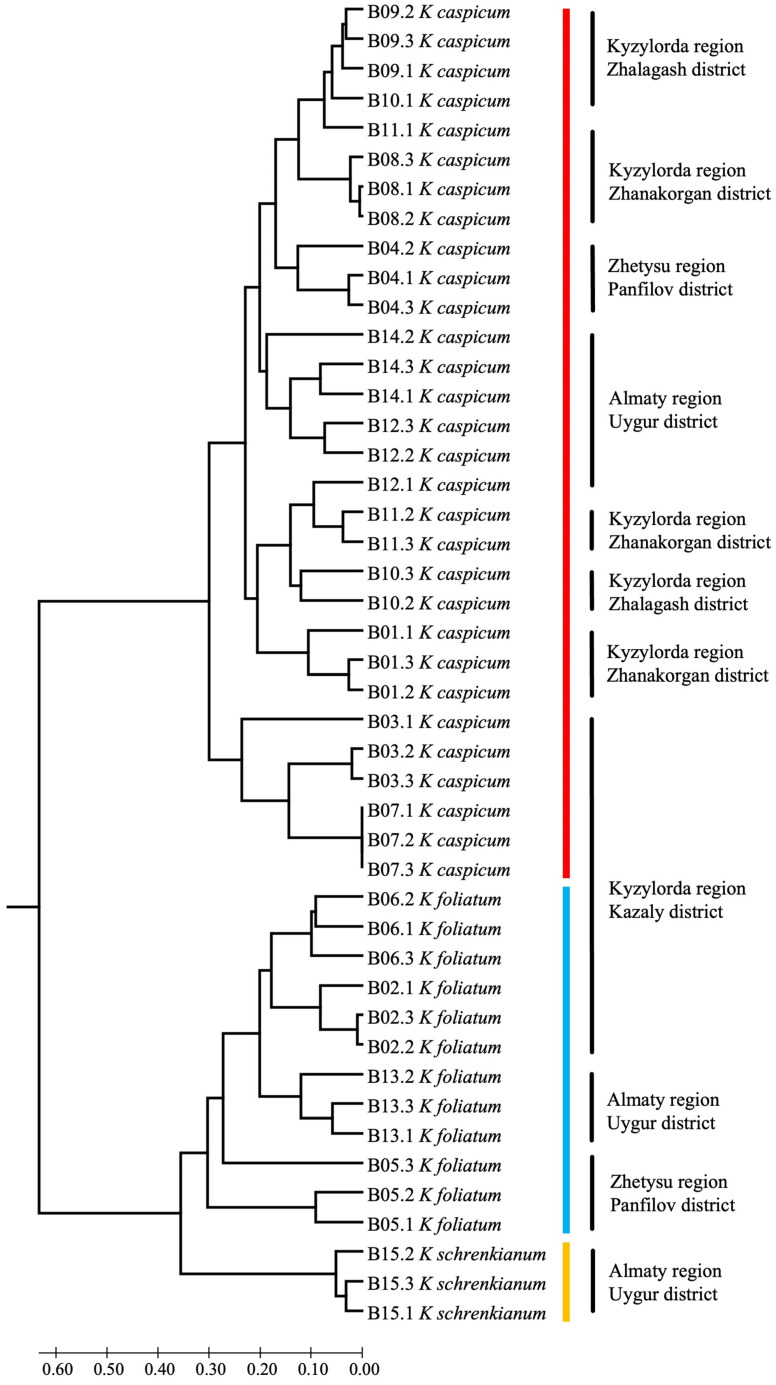
UPGMA tree based on SCoT data of the three species of the genus *Kalidium* with area and region of selected samples. For origin of the accession, see Appendix A.

**Figure 6 plants-12-02619-f006:**
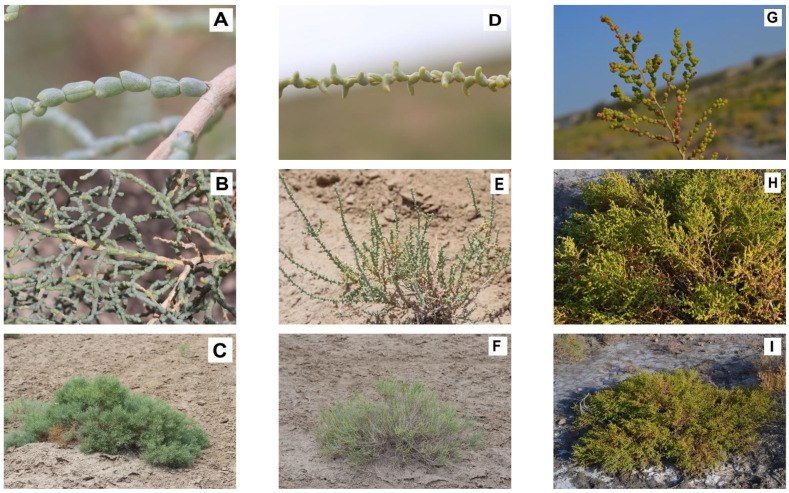
Objects of the study: (**A**–**C**)—*Kalidium caspicum* Ung.-Sternb.; (**D**–**F**)—*K. foliatum* Moq.; (**G**–**I**)—*K. schrenkianum* Bunge ex Ung.-Sternb. (all data on the collection points is provided in Appendix A).

**Table 1 plants-12-02619-t001:** DNA content in the nuclei of *Kalidium* species and expected ploidy based on cytometry data.

Species	Populations	Number of Samples Examined	DNA Content 2C ± SD, pg	Expected Ploidy Level (Literature Consensus)	Number of Chromosomes Based on Literature Data
*K. foliatum*	B02, B05, B06	9	2.259 ± 0.023	2×	18 [35,36,37,38,39,40]
*K. caspicum*	B03, B04, B07	9	2.981 ± 0.149	2×	18, 36 [41,42]
	B01, B08, B09, B10, B11	11	5.993 ± 0.139	4×	
*K. capsicum × K. foliatum*	B07 (putative hybrid)	4	2.616	2×	-

**Table 2 plants-12-02619-t002:** Ploidy data of *Kallidium* species by external standardization without changing the cytometer settings.

Pop.	Species	Average Fluorescence Value of the Peak	Index	Expected Ploidy Level
B01	*K. caspicum*	2,510,351	2.0	4×
B02	*K. foliatum*	1,091,471	1.0	2×
B03	*K. caspicum*	1,234,585	1.0	2×
B04	*K. caspicum*	1,280,461	1.0	2×
B05	*K. foliatum*	1,137,314	1.0	2×
B06	*K. foliatum*	1,159,205	1.1	2×
B07	*K. caspicum*	1,257,884	1.0	2×
B08	*K. caspicum*	2,575,620	2.1	4×
B09	*K. caspicum*.	2,535,060	2.1	4×
B10	*K. caspicum*	2,499,938	2.0	4×
B11	*K. caspicum*	2,663,129	2.2	4×

## Data Availability

All the data are presented in figures, tables, and Appendix A.

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
