# Peer review of "Phylogeny and Flow Cytometry of the Genus Kalidium Moq. (Amaranthaceae s.l.) in Kazakhstan"

_plants, 2023, doi:10.3390/plants12142619_

Round 1

Reviewer 1 Report

For the methodology, the cited paper corresponding to solving the 2C estimation and index is outdated. Similarly, the ploidy level determination using the index formula provided needs to be confirmed. For the results, the tables do not adhere to the journal's format. Likewise, the format for the citation is also bothersome and does not follow the journal's format. 

The introduction, results, and conclusions must be improved as the paragraphs have no coherence. In the conclusion part, the way it was stated is a bit confusing as well as the structure of the paragraphs is unusual. 

Reviewer 2 Report

After careful consideration and review of the manuscript (Phylogeny and flow cytometry of species of the genus Kalidium Moq. (Amaranthaceae s. lato) in Kazakhstan), I found that the results presented in the paper are not appropriate for publication.

I feel that the research presented in the manuscript does not contribute significantly to the existing body of knowledge in the field and I must reject your publication. See comments in the attached document but basically, the authors should increase the sampling to all species of the genus, clarify the methodology and substantially improve the discussion.

In my opinion, the manuscript can be greatly improved and has serious deficiencies. Moreover, I would like to request that you make some improvements to the figures presented.

Author Response

Please see in attachment

Reviewer 3 Report

All of my comments are presented in the attached pdf version of the manuscript

-

Author Response

Please see in attachment

Round 2

Reviewer 2 Report

Authors have improved the previous version of the manuscript following the recommendations of the reviewers.

Authors have improved the previous version of the manuscript following the recommendations of the reviewers.